# Formulation of SARS-CoV-2 Spike Protein with CpG Oligodeoxynucleotides and Squalene Nanoparticles Modulates Immunological Aspects Following Intranasal Delivery

**DOI:** 10.3390/pharmaceutics14112539

**Published:** 2022-11-21

**Authors:** Hui-Min Ho, Chiung-Yi Huang, Chung-Hsiang Yang, Shih-Jen Liu, Hsin-Wei Chen, Guann-Yi Yu, Jen-Kun Chen, Tsung-Hsien Chuang, Ming-Hsi Huang

**Affiliations:** 1National Institute of Infectious Diseases and Vaccinology, National Health Research Institutes, Miaoli 35053, Taiwan; 2Graduate Institute of Biomedical Sciences, China Medical University, Taichung 40402, Taiwan; 3Graduate Institute of Medicine, Kaohsiung Medical University, Kaohsiung 80708, Taiwan; 4Institute of Biomedical Engineering and Nanomedicine, National Health Research Institutes, Miaoli 35053, Taiwan; 5Immunology Research Center, National Health Research Institutes, Miaoli 35053, Taiwan

**Keywords:** COVID-19 vaccine, CpG oligodeoxynucleotides, mucosal adjuvant, nanoparticles, nasal spray delivery

## Abstract

Nasal spray vaccination is viewed as a promising strategy for inducing both mucosal and systemic protection against respiratory SARS-CoV-2 coronavirus. Toward this goal, a safe and efficacious mucosal adjuvant is necessary for the transportation of the antigen across the mucosal membrane and antigen recognition by the mucosal immune system to generate broad-spectrum immune responses. This study describes the immunological aspects of SARS-CoV-2 spike (S)-protein after being formulated with CpG oligodeoxynucleotides (ODNs) and squalene nanoparticles (termed PELC). Following intranasal delivery in mice, higher expression levels of major histocompatibility complex (MHC) class II and costimulatory molecules CD40 and CD86 on CD11c^+^ cells were observed at the draining superficial cervical lymph nodes in the CpG-formulated S protein group compared with those vaccinated with S protein alone. Subsequently, the activated antigen-presenting cells downstream modulated the cytokine secretion profiles and expanded the cytotoxic T lymphocyte activity of S protein-restimulated splenocytes. Interestingly, the presence of PELC synergistically enhanced cell-mediated immunity and diminished individual differences in S protein-specific immunogenicity. Regarding humoral responses, the mice vaccinated with the PELC:CpG-formulated S protein promoted the production of S protein-specific IgG in serum samples and IgA in nasal and bronchoalveolar lavage fluids. These results indicate that PELC:CpG is a potential mucosal adjuvant that promotes mucosal/systemic immune responses and cell-mediated immunity, a feature that has implications for the development of a nasal spray vaccine against COVID-19.

## 1. Introduction

The severe acute respiratory syndrome coronavirus 2 (SARS-CoV-2) that appeared in 2019 is causing a highly infectious respiratory disease: COVID-19 [1]. Vaccination is considered a cost-effective strategy for achieving protective immunity against SARS-CoV-2 [1,2,3]. Effective protection is mediated by systemic immunity through recognition performed by neutralizing antibodies against viral infection and transmission; in addition, vaccine-based protection from severe disease is mediated by cellular immunity to eliminate virus-infected cells [4,5]. The SARS-CoV-2 coronavirus is spread predominantly through airborne transmission or droplet transmission during coughing or sneezing from infectious patients [6]. Nasal spray vaccination is viewed as a promising strategy for inducing both mucosal and systemic protection against respiratory tract infection and virus transmission [1,2]. However, the current approved/authorized COVID-19 vaccines are mostly administered via the intramuscular route, which fails to provide protection against SARS-CoV-2 replication in the nasopharynx [1]. Due to the presence of membrane barriers, vaccine antigens applied following mucosal vaccination generally induce miniscule immune responses [7]. To achieve effective intranasal vaccination, a safe and efficacious mucosal adjuvant is necessary for the transportation of subunit antigens across the mucosal epithelium as well as antigen recognition by the mucosal immune system to synergistically generate broad-spectrum immune responses [7].

Among the protein-based COVID-19 vaccines authorized for emergency preparedness, Nuvaxovid (also known as NVX-CoV2373) is a protein-based vaccine that consists of a trimeric full-length SARS-CoV-2 spike glycoprotein using a baculovirus/insect cell expression system and a combination adjuvant (dubbed Matrix-M) comprising *Quillaja saponaria* Molina extract in a liposomal formulation [8]. MVC-COV1901, which is a protein-based vaccine authorized in Taiwan, contains a recombinant SARS-CoV-2 S-2P protein and a combination adjuvant comprising aluminum hydroxide and CpG oligodeoxynucleotides (ODNs) [9]. Alternatively, an AS03 (oil-in-water adjuvant system containing squalene and immunostimulatory α-tocopherol)-adjuvanted COVID-19 virus-like particle vaccine has been authorized in Canada [10]. Despite the evident success of subunit protein COVID-19 vaccines, no intranasal vaccine formulated with the aforementioned adjuvants is currently available to protect against COVID-19. To achieve this goal, our research group has previously described the use of squalene nanoparticles as mucosal adjuvants that allow the antigen (using ovalbumin as a model) to boost broad-spectrum antigen-specific cellular immunity following intranasal [11] or intravaginal delivery [12]. Such adjuvant ability is mediated by a fundamental role that nanoparticles play as antigen carriers in facilitating antigen transportation across the mucosal membrane to mucosal-associated lymphoid tissue (MALT). It is well appreciated that mucosal dendritic cell-mediated antigen sampling and trafficking to draining lymph nodes is important for the induction of IgA responses [7]. Therefore, it will be very interesting to investigate whether nanoparticles can modulate the immunological aspects of SARS-CoV-2 spike protein following an intranasal vaccination and reinforce the adjuvantation strategy via the codelivery of an immunostimulatory adjuvant with an optimal formulation, thereby synergistically integrating the efficacy of each individual vaccine component.

This study aims to provide comprehensive information on the methods and composition of a safe and effective adjuvant combination system using the coadministration of CpG ODNs together with squalene nanoparticles. Two investigational adjuvants, CpG-2722 and PELC, were evaluated in this study. CpG-2722 has an ODN sequence containing GTCGTT-hexamer motifs, which trigger innate immunity via the activation of toll-like receptor (TLR)-9 and TLR-21 [13]. PELC is a squalene-based nanoparticle carrier for the manipulation of antigen uptake following an intramuscular injection [14,15] as well as the transportation of antigens across nasal mucosal membranes following an intranasal delivery [16]. The goal is to combine an immunostimulatory adjuvant (CpG-2722) and a delivery carrier (PELC) synergistically, thereby driving antigen recognition at high expression levels. Figure 1 illustrates the schematic diagrams for PELC:CpG-adjuvanted COVID-19 vaccines and the vaccination schemes. Animal studies were conducted on mice to evaluate the interactions between vaccine formulations and the immune system as well as the immunogenicity of the vaccine following intranasal vaccination. Blood, nasal lavage fluid (NLF), bronchoalveolar lavage fluid (BALF), draining lymph nodes (LNs), and spleens were collected to investigate systemic, mucosal, and cell-mediated responses. These results may lead to an optimal formulation for an intranasal vaccination route against COVID-19.

## 2. Materials and Methods

### 2.1. Materials

Chemicals and reagents, unless otherwise stated, were purchased from Sigma Chemical (St. Louis, MO, USA). Reagents and media for cell culture were purchased from Thermo Fisher Scientific, Inc. (Waltham, MA, USA). Goat anti-mouse IgG, IgG1, IgG2a, and IgA antibodies (IgG: CAT#ab6789, 2 mg/mL; IgG1: CAT#ab97240, 1 mg/mL; IgG2a: CAT#ab97245, 1 mg/mL; IgA CAT#ab97235, 1 mg/mL) for enzyme-linked immunosorbent assay (ELISA) were purchased from Abcam plc. (Cambridge, UK). Antibodies and standards for cytokine ELISA were purchased from Invitrogen, Inc. (ThermoFisher, Vienna, Austria). Antibodies used for flow cytometric analysis were purchased from BioLegend, Inc. (San Diego, CA, USA). The CpG-ODN sequence (termed CpG-2722) used was 5′-GTT GTC GTT TTT TGT GCT T-3′ with all phosphorothioate backbones and was synthesized by Invitrogen, Inc. (Thermo Fisher, Carlsbad, CA, USA) [13]. Squalene nanoparticle production in a small pilot process (30 mL per batch) was prepared by a high-shear microfluidizer operating at 10,000 psi, as previously described [15]. The milky white emulsion bulk, termed PELC, contained 2.34 g of squalene, 460 mg of Span^®^85, and 360 mg of bioresorbable polymeric emulsifier poly(ethylene glycol)-block-poly(lactide-co-ε-caprolactone) suspended in 26 mL of phosphate-buffered saline (PBS) buffer. Dynamic light-scattering data showed that PELC particles issued from microfluidizer have an average intensity diameter of 189 ± 3 nm with polydispersity index (PDI) of 0.154 ± 0.019 (mean ± STD, *n* = 3). The recombinant SARS-CoV-2 subunit S protein His Tag was expressed from human embryonic kidney 293 cells (amino acid residues 16–1213, Acro Biosystems, Beijing, China). PELC:CpG combination adjuvant was prepared by redispersing 0.2 mL of PELC emulsion stock in 3.8 mL of CpG-containing PBS solution using a rotator at 5 rpm for 1 h. Prior to vaccination, the PELC:CpG-adjuvanted vaccine was formulated by mixing PELC:CpG adjuvant 1:1 (*v*/*v*) with protein solution, resulting in 2 mg/mL squalene suspended in antigenic media. The particle analysis was conducted by a light-scattering particle size analyzer (Zetasizer Pro-Blue, Malvern Panalytical Ltd., Worcester, Worcestershire, UK).

### 2.2. Mice and Ethics Statement

Four-week-old, pathogen-free BALB/c female mice were obtained from the National Laboratory Animal Center (NLAC) of Taiwan and housed for two weeks prior to experiments. All animal studies were conducted at the laboratory animal facility of the NHRI in accordance with the established institutional guidelines and the approval protocols from the IACUC review board of NHRI (NHRI-IACUC-109060-A).

### 2.3. Vaccination Schedule

When performing intranasal vaccination, mice were anesthetized with isoflurane at a maintenance concentration of 2.5% under oxygen atmosphere. To compare the mucosal adjuvant potency of CpG-2722 and PELC, five mice per group were administered—using Pipetman^®^ single channel P100L (Gilson S.A.S., Villiers-le-Bel, France) in both nostrils—with a total of 30 μL of the designed vaccine, either 20 μg of S protein in PBS, PELC nanoparticles (60 μg/dose of squalene), 5 μg/dose of CpG 2722, or their combination (Figure 1). The antigen dosage and the instillation volume were given according to a previous study on intranasal influenza subunit vaccine [16]. At week 2 and week 4, all mice were treated in the same manner with the same vaccine formulations as the priming vaccination. 

### 2.4. Systemic Antibody Responses

One week after the final vaccination, anti-sera were collected from blood samples to determine the systemic responses. The presence of SARS-CoV-2-specific IgG, IgG1, and IgG2a antibodies in the sera (starting dilution 1:200, followed by serial twofold serum dilutions) was determined by ELISA, as described previously [15]. Briefly, dilute recombinant SARS-CoV-2 spike proteins (0.1 μg/well) were coated in 96-well microtiter plates (Corning #9018, Kennebunk, ME, USA) with carbonate–bicarbonate buffer (Sigma, St-Louis, MO, USA). The coated plates were washed with PBS-Tween^®^20 and then blocked with 1% bovine serum albumin in PBS. After washing, diluted sera were added into each well, followed by the coloring reaction developed with horseradish peroxidase (HRP)-conjugated antibodies and tetramethylbenzidine (TMB) substrate solution. Finally, the coloring reaction was terminated by adding the stop solution (2N H_2_SO_4_). The titers were read as the maximum dilution that resulted in a 450 nm OD reading at least two times more than that of the preimmunize sera. An undetectable level was recorded as a titer equal to 100. Virus-neutralizing (VN) antibodies were tested on the basis of viral cytopathic effect of the homologous virus (hCoV-19/Taiwan/4/2020) in Vero cells [17]. Briefly, 100 TCID_50_ (50% tissue culture infectious dose) per well of the virus were incubated with two-fold-diluted mice sera at a starting dilution of 1:40. Mixtures of virus and heat-inactivated serum were transferred to monolayers of Vero cells and incubated at 37 °C and 5% CO_2_ for 4 days. VN antibody titers were calculated using the Reed–Muench method and expressed as the reciprocal of the highest dilution of serum sample that provided 50% neutralized infectivity, as previously described [15,17]. An undetectable level was recorded as a titer equal to 20. The data are represented as the individual values plus the geometric mean titer (GMT). The seroconversion rate (SCR) was designed as the ratio of mice with a more than 4-fold increase between pre-vaccination and postvaccination antibody titers.

### 2.5. Cell-Mediated and Mucosal Immunity

Two weeks after the final vaccination, lymphoid cells from the superficial cervical draining LNs were collected and then subjected to flow cytometry analysis (Attune^®^NxT Thermo Fisher Scientific Inc.) for CD11c, CD40, CD86, and MHC class II (CD11c: CAT#117330; CD40: CAT#124610; CD86: CAT#105030; MHCII: CAT#107626, BioLegend, San Diego, CA, USA). To investigate SARS-CoV-2-specific cytokine secretion responses, mouse spleen cell suspensions (5 × 10^6^ cells/mL) were harvested as a pooled sample and restimulated in the presence or absence of 5 μg/mL S protein. Interleukin (IL)-2, IL-10, and IL-13 concentrations in supernatants at 48 h and interferon (IFN)-γ and IL-5 concentrations at 72 h were analyzed by ELISA following the manufacturer’s instructions (IL-2: CAT#88-7024-88; IL-5: CAT#88-7054-88; IL-10: CAT#88-7105-88; IL-13: CAT#88-7137-88; IFN-γ: CAT#88-7314-88, ThermoFisher, Vienna, Austria). The single-cell suspensions were first incubated at 37 °C for 6 h with GolgiPlug^TM^ (1:1000; BD Biosciences, San Diego, CA, USA) to block cytokine secretion, resuspended in the anti-CD16/CD32 antibody solution (1 μg/10^6^ cells in 100 μL, mouse Fc Block^TM^, BD Biosciences, San Diego, CA, USA), and then analyzed with CD19, CD3, CD4, and CD8 surface markers (CD19: CAT#103116; CD3: CAT#100236; CD4: CAT#100526; CD8: CAT#100738, BioLegend, CA, USA). For intracellular cytokine staining (ICS) of IFN-γ, the cells were incubated with Cytofix/CytoPerm^TM^ fixation and permeabilization kit (BD Biosciences, San Diego, CA, USA) following the manufacturer’s instructions. The NLF and BALF were collected for mucosal IgA antibody detection. The NLF was determined by injection of 500 μL of cold PBS via the nasopharynx and the lavage fluid was collected at the nostrils. The BALF was washed by injection of 600 μL cold PBS via trachea into the lungs and the lavage fluid was aspirated at the trachea.

### 2.6. Statistical Analysis

The statistical analyses were performed using GraphPad Prism Software. Multiple comparison of the antibody titers between groups was calculated by two-way ANOVA on log_10_-transformed values followed by Bonferroni’s test. Comparison of cell-mediated immunity between the treatment group and the nonadjuvanted group was determined by a two-tailed unpaired *t* test. *p* < 0.05 was defined as statistically significant.

## 3. Results

### 3.1. Spike-PELC:CpG Formulations

Squalene nanoparticles (denoted PELC) with squalene oil suspended in PBS buffer were prepared using a high-shear microfluidized process. The particle size distribution and zeta-potential of the designed spike-PELC:CpG formulations were summarized in Table 1. Monodisperse nanoparticles with intensity mean sizes of 273 and 266 were observed in the groups of spike-PELC and spike-PELC:CpG, respectively. CpG was not crucial to the particle size of the formulations, probably due to the low CpG content in the vaccine formulations. In fact, CpG and spike protein were well-dissolved in the aqueous solution, although the polydispersity index was about 0.5. One should note that spike-PELC:CpG represents the spike protein and CpG being attached onto the PELC surfaces as they were loaded after the emulsification step [14]. The presence of CpG motifs shifted the zeta-potential of the vaccine formulations towards negative charges.

### 3.2. PELC:CpG Induces Protective Antibodies against SARS-CoV-2

The antigen-specific IgG antibody was elicited with a GMT of 9700 in sera from the mice vaccinated intranasally with S protein alone, as shown in Figure 2. When the same dosage of S protein was supplemented with the CpG-2722 adjuvant, the GMT of the anti-S protein IgG antibodies (44,600) was higher than those induced by the nonadjuvanted S protein. Interestingly, the presence of CpG-2722 can enhance IgG2a responses (*p* < 0.05) so as to bias the titer ratio of IgG2a/IgG1. On the other hand, we found that the presence of PELC in the vaccine formulation helped generate elevated GMT values of antigen-specific IgG, IgG1, and IgG2a antibodies; however, there were no statistically significant differences compared with the protein only group. PELC:CpG adjuvantation potentiates the ability of CpG to bias IgG2a/IgG1 polarization.

Figure 3 shows that the VN antibody responses were elicited in mice vaccinated with PELC:CpG-adjuvanted vaccines, both individually or in combination. Among these groups, the intranasal delivery of S protein alone or formulated with CpG alone or PELC alone could not provide full serological conversion against the induction of SARS-CoV-2 at the designed dosage following a prime-boost schedule. On the other hand, PELC:CpG-adjuvanted S protein induced higher VN titers than those developed from nonadjuvanted vaccines, in which five out of five mice had titers higher than 80 (seroconversion). Overall, both PELC- and CpG-adjuvanted vaccines could trigger the immune systems; however, only the mice that received vaccines adjuvanted with a PELC:CpG combination could provide 100% seroconversion and diminish individual variation within the host, indicating the merit of the adjuvantation platform technology in the development of intranasal COVID-19 vaccines.

### 3.3. PELC:CpG Drives IgA Production in BALF and NLF

Next, we attempted to assess the impact of PELC:CpG on the mucosal activity of a SARS-CoV-2 infection by evaluating IgA antibodies in the mucosal lavage fluids of the respiratory tract. As expected, the IgA titers in BALF and NLF were at a low or undetectable level for the mice intranasally vaccinated with nonadjuvanted S protein (Figure 4), indicating immune tolerance due to mucosal membrane barriers. Interestingly, a high level of IgA antibody secretion at the nasal microenvironment (NLF) was found for the mice that received the PELC:CpG-adjuvanted vaccine intranasally. The adjuvantation of PELC:CpG in the vaccine formulation aided IgA antibody secretion in the lower respiratory tract (BALF); however, there were no statistically significant differences compared with the protein only group. We also found that the presence of PELC nanoparticles in the vaccine formulations helped reduce the variance in each individual against S protein; PELC likely facilitates the transportation of protein antigens and/or CpG across the nasal mucosal membrane to the nasal MALT.

### 3.4. PELC:CpG Rephrases the Activation of Cell-Mediated Immunity

Cell-mediated immunity can not only play an important role in vaccine immunogenicity against ancestral strains from severe diseases but also provide cross-reactive protection against antibody-escape variant strains of SARS-CoV-2 [3,4,5]. Figure 5 and Appendix A show the cell surface expression and gating strategy of APCs in the draining LNs following intranasal vaccination with PELC:CpG-formulated S protein. As shown in Figure 5, the high cell surface expression of MHC II, CD40, and CD86 on CD11c^+^ cells was determined by flow cytometry for the harvested draining LN cells from the CpG- or PELC:CpG-treated mice. This finding can be attributed to the activation and migration of APCs at the injection site toward the draining LNs.

SARS-CoV-2-specific T-cell cytokine responses were investigated in the splenocytes following the restimulation of the cells in vitro with S protein. Figure 6 shows that sufficiently elevated IFN-γ cytokine secretion was detected in the supernatants collected from the mice treated with the CpG-adjuvanted vaccine, whereas the IL-5 and IL-10 cytokines were measured to be at the same or reduced level as those in the non-adjuvanted protein group. These findings suggest that a CpG adjuvant may be a potential tool that switches the T_H_1/T_H_2 immune balance to T_H_1 polarization. Alternatively, both T_H_1/T_H_2 cytokine levels detected in the splenocyte supernatants of the PELC group were higher than those of the control group. In addition, the mucosal delivery of the PELC:CpG adjuvant strengthens IFN-γ and IL-10 cytokines’ secretion immunity intrinsic to the individual CpG or PELC adjuvants. The PELC:CpG combination adjuvant is a promising choice, as it offers the possibility of the synergistic enhancement of immune responses against SARS-CoV-2.

Since IFN-γ secretion by T cells plays a critical role in removing virus-infected cells, we studied the functional activation marker IFN-γ on CD4^+^ and CD8^+^ cells. Figure 7 and Appendix A show the IFN-γ-secreting cells in antigen-specific CD4^+^/CD8^+^ T cells and the gating strategy following intranasal vaccination with the PELC:CpG-formulated S protein. As shown by the flow cytometric analysis (Figure 7), vaccination with S protein alone led to minimal antigen-specific IFN-γ activation. The positive population percentage of IFN-γ in CD4^+^ and CD8^+^ T cells was increased from 1.75 and 1.47 for the nonadjuvanted S protein to 2.53 and 1.82 for the CpG-adjuvanted S protein and to 3.06 and 2.11 for the PELC:CpG-adjuvanted S protein, respectively. These results demonstrated that CpG and PELC:CpG can drive high numbers of CD4^+^IFN-γ^+^ or CD8^+^IFN-γ^+^ cells in the spleens of vaccinated mice. 

## 4. Discussion

An efficacious vaccine against SARS-CoV-2 may contribute to protection via two arms of adaptive immunity: neutralizing antibodies from viral infection and T-cell immunity from the progression of a viral infection to the lower respiratory tract [4,5]. It is known that antibodies can recognize a handful of regions, receptor-binding domains or N-terminal domains, on spike protein and block viral entry via engagement with ACE-2 (the host cell receptor); the presence of antibodies in the nasal microenvironment provides complementary protection from viral entry and clinical progress [6]. On the other hand, T cells can destroy virus-infected cells by recognizing 8–15 amino acid short epitope peptides of the nucleocapsid or membrane proteins [4]. As a result, T-cell immunity plays a critical role in the face of SARS-CoV-2 viral variants. Recently, combination adjuvants comprising an immunostimulatory adjuvant and a delivery carrier have provided new insights into nasally sprayed, protein-based COVID-19 vaccine development [18]; however, different immunostimulatory adjuvants have identical action mechanisms as well as antigen–adjuvant interactions and thereby transform virus-infected cell susceptibility into CTL-mediated killing post-vaccination. There is an unmet medical need to launch a preclinical immunogenicity study to provide more information on PELC:CpG adjuvant interactions with immune cells and to elucidate their roles in vaccine immunogenicity following nasal spray delivery.

Our research group has previously described the use of amphiphilic bioresorbable polymers (PEGylated polyester/sorbitan polyester) as emulsifying agents to stabilize oil and water interfaces and give rise to versatile, multiple-phase emulsions for vaccine delivery [19]. Regarding the launch of a clinical study to evaluate the safety and immunogenicity of emulsion-adjuvanted vaccines, sufficient amounts of samples with consistent and reproducible properties should be prepared. Our research group scaled up the adjuvant preparedness process from the laboratory scale to a small pilot scale with a microfluidizer, thus providing lipid nanoparticles with a narrow size distribution. We previously applied this platform in developing a single-dose, intramuscular COVID-19 vaccine; the data showed that one dose of S protein with PELC works comparably to repeated doses of S protein alone for both B-cell and T-cell immunity [15]. Here, we demonstrated that PELC nanoparticles act as antigen carriers to facilitate the transportation of S protein antigen and CpG across nasal mucosal membrane to MALT and trigger vaccine immunogenicity. In the absence of CpG, PELC help S proteins to induce a strong T-cell immune responses, which is a feature suitable for the development of an efficacious COVID-19 vaccine to protect against antibody-escape SARS-CoV-2 viral variants and severe disease. However, we did not observe adjuvant potency of the PELC nanoparticles with respect to B-cell immunity when the mice were vaccinated intranasally. Interestingly, our results indicate that PELC:CpG is a promising mucosal adjuvant for inducing protective antibodies against SARS-CoV-2, to drive IgA production in BALF and NLF, and to rephrase the activation of cell-mediated immunity; in addition, the PELC:CpG combination could provide 100% seroconversion and diminish individual variation within the host. Notably, CpG ODNs have species-specific activity, which is determined by their structural features, such as the context of the CpG-hexamer motifs and their length [13]. Here, CpG-2722 contains one copy of the GTCGTT motif that effectively activates immune responses in human and mouse cells. In addition to neutralizing the antibody response, a correlation between antibody-dependent cellular cytotoxicity (ADCC) responses and COVID-19 disease development has been demonstrated; the activity that mediates ADCC may play a role in effective vaccination and therapeutic interventions against COVID-19 [20]. 

Currently, the respiratory mucosal delivery of the COVID-19 vaccine is primarily developed with viral-vector vaccine candidates [1]. To the best of the authors’ knowledge, an effective combination of CpG and squalene nanoparticles for the development of a nasal spray subunit protein antigen vaccine against COVID-19 has not been reported in the literature. Our results provide critical immunological insights into the nasal mucosal activity of the PELC:CpG adjuvant from the viewpoint of immunologists/material scientists and provide directions for designing an optimal mucosal vaccine formulation. Further studies are warranted with respect to finding an optimal PELC:CpG dosage to develop a COVID-19 nasal spray vaccine for both ADCC activity and neutralizing antibody response. Such an application requires a deepened understanding of the identification of the manufacturing parameters governing the physicochemical and immunological properties of the emulsion adjuvants issued from a small pilot-scale, high-shear fluid process.

## 5. Conclusions

Here, we propose an adjuvantation strategy to enhance and modulate the immunogenicity of a SARS-CoV-2 spike protein vaccine antigen through the intranasal route. The goal was to use the codelivery of a combination adjuvant comprising an immunostimulatory adjuvant (CpG) and a delivery carrier (PELC) synergistically to drive immune responses. Our immunogenicity studies in mice have demonstrated that CpG ODNs can skew the immune responses toward T_H_1. Furthermore, squalene nanoparticles are a potent tool for the effective delivery of CpG ODNs along with protein antigens. The COVID-19 vaccine candidates adjuvanted with the optimal formulation work toward the enrichment of the immune microenvironment in draining LNs following intranasal vaccination. The optimal combination of adjuvants can achieve successful vaccination by eliciting T_H_1-type T-cell polarization, adequate mucosal protection, and high seroconversion and are thus an effective tool for pandemic-scale vaccine preparedness.

## Figures and Tables

**Figure 1 pharmaceutics-14-02539-f001:**
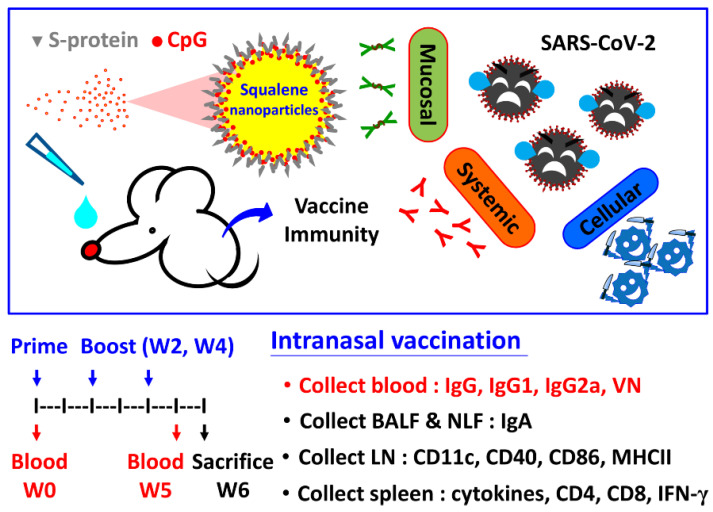
Schematic representation and immunogenicity study design of CpG- and PELC-adjuvanted COVID-19 vaccines. PELC:CpG is a combination adjuvant comprising PELC squalene nanoparticles and CpG oligodeoxynucleotides. Following intranasal delivery, PELC act as antigen carriers to facilitate the transportation of antigen and CpG across nasal mucosal membrane to mucosal-associated lymphoid tissue and trigger vaccine immunogenicity. Mice were intranasally (i.n.) administered full-length SARS-CoV-2 spike subunit protein, either with CpG, PELC, or their combination. At week 2 and week 4, all mice were boosted i.n. with the same vaccine formulations. Draining LNs were harvested to determine antigen-presenting cell (APC) activation. Blood, NLF, BALF, and spleen collections were performed to determine systemic, mucosal, and cell-mediated responses.

**Figure 2 pharmaceutics-14-02539-f002:**
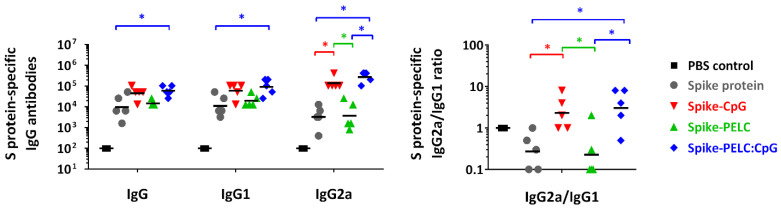
Antigen-specific antibody responses following three intranasal doses of PELC:CpG-formulated S protein in BALB/c mice. Mice (*n* = 5) were intranasally (i.n.) administered full-length SARS-CoV-2 spike subunit protein either with CpG, PELC, or their combination. At week 2 and week 4, all mice were boosted i.n. with the same vaccine formulations. The total IgG and IgG subtype antibodies in serum samples at week 5 are expressed as the individual values with the GMT. The log-transformed values of antibody titers were compared by employing ANOVA model followed by Bonferroni’s multiple comparison test. * *p* < 0.05.

**Figure 3 pharmaceutics-14-02539-f003:**
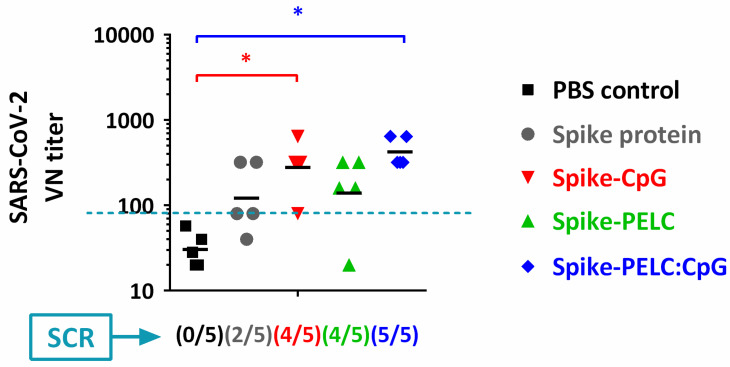
SARS-CoV-2 virus neutralizing (VN) responses following three intranasal doses of PELC:CpG-formulated S protein in BALB/c mice. Mice (*n* = 5) were intranasally (i.n.) administered full-length SARS-CoV-2 spike subunit protein, either with CpG, PELC, or their combination. At week 2 and week 4, all mice were boosted i.n. with the same vaccine formulations. The VN antibodies in serum samples at week 5 are expressed as the individual values with the GMT. The seroconversion rate (SCR) was calculated from the proportion of mice with a more than 4-fold increase between pre-vaccination and postvaccination antibody titers. The log-transformed values of antibody titers were compared by employing ANOVA model followed by Bonferroni’s multiple comparison test. * *p* < 0.05. The dashed line represents seroconversion.

**Figure 4 pharmaceutics-14-02539-f004:**
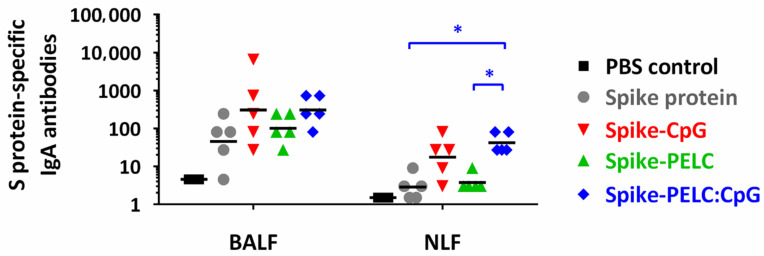
Antigen-specific mucosal response following three intranasal doses of PELC:CpG-formulated S protein in BALB/c mice. Mice (*n* = 5) were intranasally (i.n.) administered full-length SARS-CoV-2 spike subunit protein, either with CpG, PELC, or their combination. At week 2 and week 4, all mice were boosted i.n. with the same vaccine formulations. The IgA antibodies in BALF and NLF samples at week 6 are expressed as the individual values with the GMT. The log-transformed values of antibody titers were compared by employing ANOVA model followed by Bonferroni’s multiple comparison test. Starting sample dilution of 1:9 for BALF and 1:3 for NLF followed serial threefold dilutions. * *p* < 0.05.

**Figure 5 pharmaceutics-14-02539-f005:**
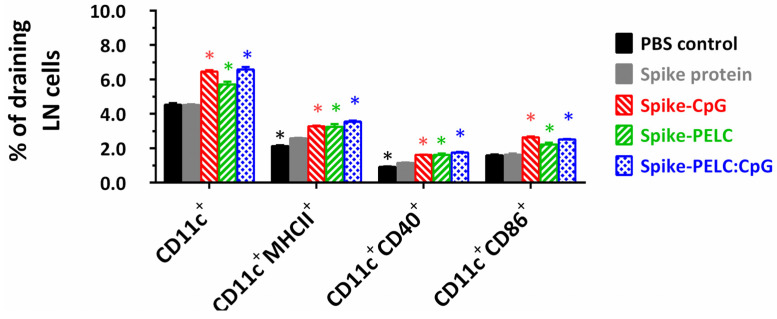
Cell surface expression of APCs in draining lymph nodes (LNs). The mice were given the designed formulations once every two weeks a total of three times. Two weeks after the final administration, the mice were sacrificed, and the cells from superficial cervical draining LNs were harvested and stained with fluorescence-labeled anti-CD40, anti-CD86, anti-MHC II, and anti-CD11c antibodies. The levels of CD40, CD86, and MHC II expression on live CD11c^+^ cells were determined by flow cytometry. The positive population percentage data are represented as the mean ± standard deviation from triplicate samples of pooled LN cells. * *p* < 0.05 in comparison with nonadjuvanted S protein group.

**Figure 6 pharmaceutics-14-02539-f006:**
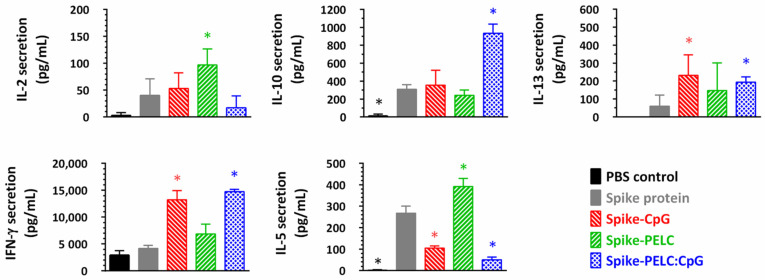
Cytokine secretion response following intranasal vaccination with PELC:CpG-formulated S protein. BALB/c mice were given three intranasal doses of SARS-CoV-2 S protein antigen. Two weeks after the final vaccination, splenocytes (5 × 10^6^ cells/mL) were harvested and then incubated in the presence of S protein (5 μg/mL). Supernatants collected from cell cultures were assessed by IFN-γ, IL-2, IL-5, IL-10, and IL-13 ELISA. The data are presented as cytokine release in the presence of S protein minus those released in the absence of S protein and expressed as the mean ± standard deviation of triplicate cultures of pooled spleen samples. * *p* < 0.05 in comparison with nonadjuvanted S protein group.

**Figure 7 pharmaceutics-14-02539-f007:**
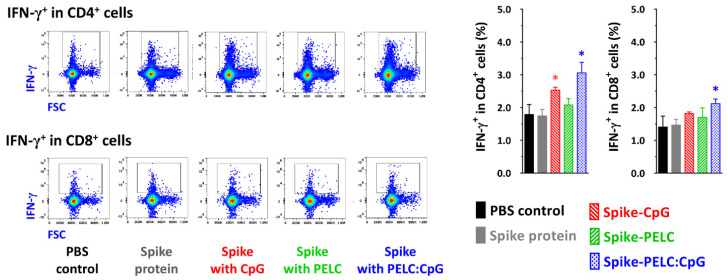
IFN-γ-secreting cells in antigen-specific CD4^+^/CD8^+^ T cells following intranasal vaccination with PELC:CpG-formulated S protein. BALB/c mice were given three intranasal doses of SARS-CoV-2 S protein antigen. Two weeks after the final vaccination, the mice were sacrificed, and splenocytes (5 × 10^6^ cells/mL) were harvested and then incubated with S protein (5 μg/mL) for 24 h. Splenic CD19-excluded CD3^+^ cells were analyzed by flow cytometry to determine the levels of IFN-γ expression in CD4^+^ or CD8^+^ cells. The positive population percentage data within the rectangle gates are represented as the mean ± standard deviation from triplicate cultures of pool murine spleen samples. * *p* < 0.05 in comparison with nonadjuvanted S protein group.

**Table 1 pharmaceutics-14-02539-t001:** Particle analysis of the designed spike-PELC:CpG formulations by light-scattering technology ^(a)^.

Sample ID	Z-Avg (nm) ^(b)^	PDI ^(c)^	Zeta-Potential (mV)
PBS	N.D.	N.D.	−22.9 ± 10.5
Spike protein	105 ± 4	0.541 ± 0.036	−5.9 ± 2.5
Spike-CpG	110 ± 3	0.502 ± 0.019	−14.2 ± 10.0
Spike-PELC	273 ± 0	0.139 ± 0.008	−10.6 ± 1.9
Spike-PELC:CpG	266 ± 4	0.176 ± 0.014	−12.6 ± 0.6

^(a)^ The data are represented as means with standard deviation (STD) of three samples. ^(b)^ Z-avg: Z-average size mean by intensity. ^(c)^ PDI: polydispersity index. N.D.: nondetectable.

## Data Availability

The data presented in this study are available in Appendix A.

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
