# Peer review of "Formulation of SARS-CoV-2 Spike Protein with CpG Oligodeoxynucleotides and Squalene Nanoparticles Modulates Immunological Aspects Following Intranasal Delivery"

_pharmaceutics, 2022, doi:10.3390/pharmaceutics14112539_

Round 1

Reviewer 1 Report

The manuscript pharmaceutics-2027005 entitled Formulation of SARS-CoV-2 Spike Protein with CpG Oligodeoxynucleotides and Squalene Nanoparticles Modulates Immunological Aspects Following Intranasal Delivery by Hui-Min Ho , Chiung-Yi Huang , Chung-Hsiang Yang , Shih-Jen Liu , Hsin-Wei Chen , Guann-Yi Yu , Jen-Kun Chen , Tsung-Hsien Chuang , Ming-Hsi Huang described the vaccine immune response in mice induced by a intranasally administered formulation of the SARS-CoV2’s spike protein with squalene nanoparticle adjuvanted with CpG. While the results are interesting in the field, they are over-interpreted.

Major corrections of the text are required.

- Line 75 : reference 10 is not the good one. It does not make reference to intranasally administered squalene-ovalbumin. Please correct by 11 then shift the following.

- paragraph 2.5 : the full reference of all the components should be described (antibody, ELISA kits…)

- paragraph 2.3 : Are the mice awake or asleep when intranasally administered ? It has involvement on the ability for the formulation to be correctly captured by the nasal epithelium. The volume administered is also important. The authors should discuss that point

- A major point relates to the description and the characterization of the squalene(PELC):CpG:Spike protein formulation. Authors have only referred to previous studies with Squalene(PELC??):Spike or PELC:CpG:other antigens but no clear data is presented on the PELC:CpG:Spike.

Authors must further describe their formulation. Only the size is indicated in the methods section.

- How long is the stability (colloidal, release of CpG and Spike protein) of the PELC:CpG:formulation ?

- What is the mode of the hydrodynamic diameter (intensity, number, Z-Average)? What is the PDI ?

- What is the diluent of the measurement?

- What is the zeta potential of the formulation?

- How do the CpG and the Spike protein influence these parameters?

- How do the components interact with each other to create the formulation?

- How to confirm that Spike protein is associated to squalene NP?

- The figures show that CpG:Spike gives equivalent results to PELC:CpG:Spike (no significant statistical difference, no clear biological difference), but the authors always explain that PELC:CpG:Spike is better than CpG:Spike (for example : lines 209+, 228+, 249+…).

The only clear difference between those two formulations are IL2, IL10 (fig 6) and IFN+CD8+ (fig 7).

On Figure 3, authors tell that only the group PELC:CpG:Spike show 100% seroconversion but its is also written 5/5 for the CpG:Spike group.

Authors must re-write or prove all these statements.

Author Response

Point 1: Line 75 : reference 10 is not the good one. It does not make reference to intranasally administered squalene-ovalbumin. Please correct by 11 then shift the following.

Response 1: The authors thank the reviewer for correcting this error; reference 10 was corrected by 11 then shifted the following.

Point 2: paragraph 2.5 : the full reference of all the components should be described (antibody, ELISA kits…).

Response 2: The authors concurred with the Reviewer's suggestion giving the full reference of all the components.

Point 3: paragraph 2.3 : Are the mice awake or asleep when intranasally administered ? It has involvement on the ability for the formulation to be correctly captured by the nasal epithelium. The volume administered is also important. The authors should discuss that point.

Response 3: The mice were anesthetized when performing intranasal vaccination in accordance with the guidelines of the Laboratory Animal Center of NHRI.

  • A statement was added, "When performing intranasal vaccination, mice were anesthetized with isoflurane at a maintenance concentration of 2.5% under oxygen."
  • The statement, "The antigen dosage was given according to a previous study for intranasal influenza subunit vaccine [16]." was replaced as, "The antigen dosage and instillation volume were given according to a previous study for intranasal influenza subunit vaccine [16]."

Point 4: A major point relates to the description and the characterization of the squalene(PELC):CpG:Spike protein formulation. Authors have only referred to previous studies with Squalene(PELC??):Spike or PELC:CpG:other antigens but no clear data is presented on the PELC:CpG:Spike. Authors must further describe their formulation. Only the size is indicated in the methods section.

How long is the stability (colloidal, release of CpG and Spike protein) of the PELC:CpG formulation ?.

What is the mode of the hydrodynamic diameter (intensity, number, Z-Average)? What is the PDI ?

- What is the diluent of the measurement?

- What is the zeta potential of the formulation?

- How do the CpG and the Spike protein influence these parameters?

- How do the components interact with each other to create the formulation?

- How to confirm that Spike protein is associated to squalene NP?.

Response 4: This manuscript focuses on the immunogenicity of SARS-CoV-2 S-protein after being formulated with a combination adjuvant PELC:CpG following intranasal delivery in mice. However, the authors agree with the Reviewer's comment, a complete understanding of how the adjuvant mechanisms linked with the structure-function of PELC:CpG is underway to extend platform technology for prophylactic vaccine against emerging infectious diseases and therapeutic vaccine against cancers. The authors concurred with the Reviewer's suggestion describing in detail the formulations in the Section 3.1.

3.1. Spike-PELC:CpG Formulations

Squalene nanoparticles (named as PELC) with squalene oil suspended in PBS buffer could be prepared under high-shear microfluidized process. The particle size distribution and zeta-potential of the designed spike-PELC:CpG formulations were summarized in Table 1. Monodisperse nanoparticles with the intensity mean size of 273 and 266 were observed in the groups of spike-PELC and spike-PELC:CpG, respectively. CpG is not crucial to the particle size of the formulations, probably due to small amount of CpG content in the vaccine formulations. In fact, CpG and spike protein are well dissolved in the aqueous solution, although the polydispersity index is about 0.5. One should note that spike-PELC:CpG represents the spike protein and CpG being attached onto PELC surface as they were loaded after the emulsification step [14]. The presence of CpG motives shifted the zeta-potential of the vaccine formulations towards negative charges.

Table 1. Particle analysis of the designed spike-PELC:CpG formulations by light scattering technology a)

Sample ID

Z-avg (nm) b)

PDI c)

Zeta-potential (mV)

PBS

N.D.

N.D.

-22.9±10.5

Spike protein

105±4

0.541±0.036

-5.9±2.5

Spike-CpG

110±3

0.502±0.019

-14.2±10.0

Spike-PELC

273±0

0.139±0.008

-10.6±1.9

Spike-PELC:CpG

266±4

0.176±0.014

-12.6±0.6

  1. a) The data were represented as the mean with standard deviation (STD) of three samples.
  2. b) Z-avg: Z-average size mean by intensity. c) PDI: polydispersity index. N.D.: nondetectable.

Point 5: The figures show that CpG:Spike gives equivalent results to PELC:CpG:Spike (no significant statistical difference, no clear biological difference), but the authors always explain that PELC:CpG:Spike is better than CpG:Spike (for example : lines 209+, 228+, 249+…). The only clear difference between those two formulations are IL2, IL10 (fig 6) and IFN+CD8+ (fig 7).

Response 5: In this study, we found that PELC:CpG keeps the merit of CpG to bias the IgG2a/IgG1 polarization and diminish individual variation in VN and IgA. However, the authors agree with the reviewer's comment that the statement is not appropriate in the manuscript. Towards this, these statements were revised.

Line 235, the statement, "Interestingly, the presence of CpG-2722 (CpG alone or PELC:CpG combination) can enhance IgG2a responses (p<0.05) so as to bias the titer ratio of IgG2a/IgG1. On the other hand, we found that the presence of PELC in the vaccine formulation helped generate elevated GMT value of antigen-specific IgG, IgG1, and IgG2a antibodies; however, there were no statistically significant differences compared with protein alone group." was replaced as,

"Interestingly, the presence of CpG-2722 can enhance IgG2a responses (p<0.05) so as to bias the titer ratio of IgG2a/IgG1. On the other hand, we found that the presence of PELC in the vaccine formulation helped generate elevated GMT value of antigen-specific IgG, IgG1, and IgG2a antibodies; however, there were no statistically significant differences compared with protein alone group. PELC:CpG adjuvantation comprises the merit of CpG to bias the IgG2a/IgG1 polarization."

Line 250, the statement, "Among these groups, intranasal delivery of S protein alone or formulated with PELC could not provide full serological conversion against SARS-CoV-2 induction at the designed dosage following prime-boost schedule. On the other hand, CpG- or PELC:CpG-adjuvanted S protein induced higher VN titers than those developed from nonadjuvanted vaccines in which five out of five mice had titers higher than 80 (seroconversion). Overall, both PELC- and CpG-adjuvanted vaccines could trigger the immune systems; however, only the mice that received vaccines adjuvanted with PELC:CpG combination could provide 100% seroconversion and diminish individual variation among host, indicating the merit of the adjuvantation platform technology in the development of intranasal COVID-19 vaccines." was replaced as,

"Among these groups, intranasal delivery of S protein alone or formulated with CpG alone or PELC alone could not provide full serological conversion against SARS-CoV-2 induction at the designed dosage following prime-boost schedule. On the other hand, PELC:CpG-adjuvanted S protein induced higher VN titers than those developed from nonadjuvanted vaccines in which five out of five mice had titers higher than 80 (seroconversion). Overall, both PELC- and CpG-adjuvanted vaccines could trigger the immune systems; however, only the mice that received vaccines adjuvanted with PELC:CpG combination could provide 100% seroconversion and diminish individual variation among host, indicating the merit of the adjuvantation platform technology in the development of intranasal COVID-19 vaccines."

Line 275, the statement, "Interestingly, a high level of IgA antibody secretion at the nasal microenvironment (NLF) and the lower respiratory tract (BALF) was found for the mice that received intranasally PELC:CpG-adjuvanted vaccine." was replaced as,

"Interestingly, a high level of IgA antibody secretion at the nasal microenvironment (NLF) was found for the mice that received intranasally PELC:CpG-adjuvanted vaccine. The adjuvantation of PELC:CpG in the vaccine formulation helped IgA antibody secretion at the lower respiratory tract (BALF); however, there were no statistically significant differences compared with protein alone group."

Point 6: On Figure 3, authors tell that only the group PELC:CpG:Spike show 100% seroconversion but its is also written 5/5 for the CpG:Spike group.

Response 6: The authors agree with the reviewer's comment that the statement is not appropriate in the manuscript. Towards this, the statement, "The seroconversion rate (SCR) was designed as the ratio of mice with a minimum 4-fold increase between prevaccination and postvaccination antibody titers." was rewritten as "The seroconversion rate (SCR) was designed as the ratio of mice with more than 4-fold increase between prevaccination and postvaccination antibody titers." Based on this definition, SCR is 2/5 for the Spike group and 4/5 for the CpG:Spike group.

Reviewer 2 Report

Formulation of SARS-CoV-2 Spike Protein with CpG Oligodeoxynucleotides and Squalene Nanoparticles Modulates Immunological Aspects Following Intranasal Deliver is a very interesting study by Ming-Hsi Huang et al.,

1.     Results obtained from DLS must be incorporated in the revised manuscript to understand better readers regarding the size, charge, and PDI of particles used in the study.

2.     Add about lymph nodes and how they work to generate immunity in the introduction section.

Author Response

Point 1: Results obtained from DLS must be incorporated in the revised manuscript to understand better readers regarding the size, charge, and PDI of particles used in the study.

Response 1: The authors concurred with the Reviewer's suggestion describing in detail the formulations in the Section 3.1.

3.1. Spike-PELC:CpG Formulations

Squalene nanoparticles (named as PELC) with squalene oil suspended in PBS buffer could be prepared under high-shear microfluidized process. The particle size distribution and zeta-potential of the designed spike-PELC:CpG formulations were summarized in Table 1. Monodisperse nanoparticles with the intensity mean size of 273 and 266 were observed in the groups of spike-PELC and spike-PELC:CpG, respectively. CpG is not crucial to the particle size of the formulations, probably due to small amount of CpG content in the vaccine formulations. In fact, CpG and spike protein are well dissolved in the aqueous solution, although the polydispersity index is about 0.5. One should note that spike-PELC:CpG represents the spike protein and CpG being attached onto PELC surface as they were loaded after the emulsification step [14]. The presence of CpG motives shifted the zeta-potential of the vaccine formulations towards negative charges.

Table 1. Particle analysis of the designed spike-PELC:CpG formulations by light scattering technology a)

Sample ID

Z-avg (nm) b)

PDI c)

Zeta-potential (mV)

PBS

N.D.

N.D.

-22.9±10.5

Spike protein

105±4

0.541±0.036

-5.9±2.5

Spike-CpG

110±3

0.502±0.019

-14.2±10.0

Spike-PELC

273±0

0.139±0.008

-10.6±1.9

Spike-PELC:CpG

266±4

0.176±0.014

-12.6±0.6

  1. a) The data were represented as the mean with standard deviation (STD) of three samples.
  2. b) Z-avg: Z-average size mean by intensity. c) PDI: polydispersity index. N.D.: nondetectable.

Point 2: Add about lymph nodes and how they work to generate immunity in the introduction section.

Response 2: The authors concurred with the Reviewer's suggestion adding a statement, "It is well appreciated that the mucosal dendritic cell-mediated antigen sampling and trafficking to draining lymph nodes is important for induction of IgA responses [7]."
